# Color Measurement in the Corrosivity Assessment of Museums, Archives, and Churches

**DOI:** 10.3390/ma16010226

**Published:** 2022-12-26

**Authors:** Tereza Boháčková, Milan Kouřil, Kristýna Charlotte Strachotová, Kateřina Kreislová, Pavlína Fialová, Jan Švadlena, Tomáš Prošek

**Affiliations:** 1Department of Metals and Corrosion Engineering, Faculty of Chemical Technology, University of Chemistry and Technology Prague, Technická 5, 166 28 Prague, Czech Republic; 2SVÚOM s.r.o., U Měšťanského Pivovaru 934/4, 170 00 Prague, Czech Republic; 3Department of Metallic Construction Materials, Technopark Kralupy of the University of Chemistry and Technology, Náměstí G. Karse 7, 278 01 Kralupy nad Vltavou, Czech Republic

**Keywords:** colorimetric measurement, volatile organic acids, corrosion coupons, indoor corrosivity

## Abstract

Indoors, volatile organic acids can play an important role in the degradation process of many materials. Considering this fact, metal corrosion coupons of copper, silver, lead, and zinc were exposed to different climatic conditions of 18 locations for 3, 12, and 30 months, and their corrosion rates were evaluated based on mass loss, as recommended by the ISO 11844 standard. The corrosion rates were compared with in situ colorimetric measurements to validate the colorimetry as a simple tool for estimating the corrosivity of an environment. The results have shown good correlation between the methods for two metals: silver and lead, confirming the possibility of non-destructive monitoring of their corrosion by measurement of color changes.

## 1. Introduction

Although indoor environments have reduced concentrations of pollutants compared to outdoor conditions, there are aggressive substances whose concentrations increase indoors, especially if air circulation is restricted [1]. They include volatile organic acids such as acetic or formic acid, to which many materials are sensitive, among them metals, most notably lead [2]. Therefore, it is desirable to deal with the issue of environmental aggressiveness indoors as well, where collections of art objects are stored or displayed.

In the context of a low-aggressive condition, it is possible to follow the ISO 11844 standard. According to the standard, corrosivity of an environment can be determined using metal coupons of silver, copper, iron, zinc, and lead that are exposed to a monitored location for a year [3]. Subsequently, the corrosive aggressiveness of the environment is evaluated based on mass changes of coupons. This method of assessment is limiting in several respects. The first is the long waiting time to obtain results, while environmental conditions and potential emission sources can change, typically during temporary exhibitions. The second limitation is the method of evaluation itself. In low-aggressive conditions, mass changes of metal coupons are small, requiring the use of precise ultra-microbalances to detect the changes. However, they are expensive, and their operation requires stable conditions without possible sources of vibration.

Another way of evaluating the aggressiveness, also mentioned in the standard [4], is the measurement of levels of air pollutants. Measuring concentrations of airborne pollutants requires accurate and costly equipment and trained operators. Alternatively, commercially available samplers offer an indicative determination of pollutant concentrations based on a color reaction and the associated calibration scale but are often set to a higher detection limit than necessary. In addition, other pollutants may interfere and influence the color [5].

Institutions that store collected objects and their staff usually do not need to know the exact concentrations of hazardous substances, but they must be able to respond flexibly to changing conditions and protect collection objects using available equipment and methods. An available instrument that museums, archives, or libraries usually have or have access to it is a spectrophotometer for measuring color changes. It is widely used to monitor, for example, color changes caused by an application of cleaning and protective agents [6,7,8], degradation of materials [9,10,11,12,13], or patina formation [14,15,16]. Both the total color difference and the individual color parameter can be evaluated. This paper discusses the use of colorimetric measurement as a simple detection method for environmental corrosivity and compares it with the ISO 11844 procedures of mass loss of metal coupons exposed for 3–30 months in 18 locations, considering results recently published elsewhere [17].

## 2. Materials and Methods

### 2.1. Overview of Locations and Their Climatic Parameters

Eighteen locations with different storage conditions in which metal objects are stored or which contain VOC-emitting materials were selected. These environments can be divided into a few groups. The first group consisted of depositories (locations 6–7, 13) or archives (locations 8–10, 17) where conditions are controlled: in the case of locations 6 and 7, it is only humidity and temperature, while in the case of locations 8–10 and 17, it is air circulation and filtration, too. The second type was exhibition spaces in libraries or museums (locations 1, 3, 5) where the temperature is adapted to the presence of visitors and where there are small fluctuations in temperature and humidity during the year. These spaces contain many potential sources of VOCs, typically wooden floors and furniture. Wooden cabinets or showcases also represent an environment with limited air exchange, and therefore, these cryptoclimates were selected as a third type (locations 2, 4, and 11). Conditions of locations 12 and 18 also correspond to the cryptoclimate of the enclosed space, but the aggressiveness of the environment was increased by the placement of cellulose acetate films. The last group consisted of churches (locations 14–16), i.e., unheated spaces where due to leakage of roofs, windows, or doors, there is no control over the indoor conditions, and temperature and humidity vary significantly both during the day and the year. More detailed descriptions of the sites 1–16 can be found in [17].

Temperature and humidity in locations were monitored using the Testo 174H loggers (Testo SE & Co., KGaA, Lenzkirch, Germany). The content of acetic acid in the air was measured using passive samplers produced by the authors and analyzed by means of ion chromatography, as described in [18]. The acetic acid concentration was monitored for a year because of a strong dependence of its emission on atmospheric conditions [19]. The samplers were changed at monthly intervals and the concentration was the average from three measurements. Table 1 summarizes the annual average values, minima, and maxima of temperature, relative humidity, and acetic acid concentration. The table does not include values for locations 9 and 10, for which coupons were placed in archive boxes located in the depository marked as location 8. Similarly, temperature and humidity values were not recorded for location 18: this was a wooden box with an acetate film stored in location 17. The small space made it impossible to measure acetic acid levels in these locations as well as in location 12.

### 2.2. Preparation of Coupons and Their Exposure

Based on the ISO 11844 standard, four metals were selected for exposure: copper (1 mm sheet of a 99.5% purity, Ferona a.s., Prague, Czech Republic), silver (0.5 mm sheet of a 99.99% purity, Safina a.s., Prague, Czech Republic), lead (0.5 mm sheet of a 99.97% purity, Kovohutě Příbram nástupnická, a.s., Příbram, Czech Republic), and zinc (1 mm sheet of a 99.5% purity, Ferona a.s., Prague, Czech Republic). Two sets of six coupons were prepared for each metal, and their dimensions of 10 × 25 mm were chosen due to the limited weighing capacity of ultra-microbalances, which is 2 g. Before exposing, they were weighed, and most of them hung on a plastic rack, as recommended by the standard. Only coupons in locations 9, 10, 12, and 18 laid horizontally due to the small space. Exposure took place in 2019–2022. As recommended by the standard, after 12 and 30 months, one set of coupons was weighed, and mass loss was determined. Color changes were measured before the start of exposure and after 1, 3, 6, 12, 24, and 30 months. During the long-term exposures, smaller coupons (10 × 10 mm) were made of copper, silver, and lead sheets, and two coupons of each metal were placed in the location for a period of three months. The three-month exposure was additionally carried out to compare the mass changes with the color changes that were already significant after the first months of exposure.

### 2.3. Evaluation Methods

Mass loss of metal coupons was evaluated in accordance with ISO 11844-2, which determines the weighing of ultra-microbalances (UYA 2.4Y Ultra-microbalance, Radwag, Radom, Poland) in relation to a reference balance standard (stainless-steel of similar mass), and specifies formulas for the calculation of corrosion rates, as well as a one-step pickling process. Since the pickling solution can also attack the uncorroded metal surface, the mass loss of a comparative unexposed sample pickled at the same time was subtracted from that of the exposed sample to eliminate this effect.

The composition of the pickling baths complied with the requirements of the standard: all baths operated at laboratory temperature and pickling usually took one minute. Only for some coupons exposed for 30-month exposures was it necessary to extend the pickling time to double or triple, but in this case, the comparative unexposed sample was pickled for the same length of time. After pickling, the samples were rinsed in demineralized water and ethanol and then dried. To calculate corrosion rates, the area of both sides of the coupons was considered, neglecting the edges, and holes were drilled for fixing the specimens in the rack. With a maximum sample thickness of 0.5 mm, its omission from calculation did not exceed 10% of the total value.

During exposure, color changes were regularly monitored on 2 × 1 cm samples using a portable Konica Minolta CM-700d spectrophotometer with a 3 mm aperture (Konica Minolta GmbH, Munich, Germany). Two measurements in the CIELAB color space were performed from each side of the coupon in both specular component-excluded and -included (SCE and SCI). The total color difference was calculated from the changes in measured parameters a^*^, b^*^, and L^*^. Since the results of the two modes did not differ much and showed the same trends, only the results of the SCI mode are described in this paper. Changes in an appearance of metallic coupons were also visually monitored under the Olympus SZX10 stereomicroscope (Olympus Europa SE & Co., KG, Hamburg, Germany).

## 3. Results

### 3.1. Color Changes of Coupons

Results of colorimetric measurements at different stages of exposures showed gradual changes in color, which have a general tendency to slow down with time. However, different trends could be observed within individual materials. For silver, total color differences varied significantly during exposures (Figure 1), so the differences between the first and last months were multiple. The biggest changes did not occur in environments with high acetic acid concentrations; in fact, the values from exposures with acetate films (locations 12 and 18) were among the lowest. Probably, limited air exchange and access to other pollutants caused the color changes to be comparable to those from the controlled conditions of location 8. Minimal changes in appearance, with only slight yellowing of the coupons, were also observable under a stereomicroscope (Figure 2). The low sensitivity of silver to acetic acid was also evident by comparing results from locations 1 and 2 or from locations 2 and 4. The lower of the two numbers represents exposure in an open area, while the higher are coupons from a wooden cabinet inside locations 1 and 3. In the latter case, the color changes were always smaller, even though the acid concentration was at least twice as high. One of the most significant changes in color occurred on silver coupons from location 15 (Figure 3), where large fluctuations in temperature and relative humidity, air ventilation, and remnants of bird activity promoted the corrosion processes.

Results of the colorimetric measurement on copper coupons (Figure 4) showed similar trends to the silver samples, namely an increasing color change throughout the exposure time of most coupons as well as a negligible effect of the acetic acid concentration. Likewise, the greatest changes in appearance were in locations 1 and 15, but visually, the coupons differed considerably more (Figure 5), and it was not the only discrepancy. There is no clear explanation for the largest total color differences in the first few months of exposure in locations 3, 4, 6, and 7, nor did the appearance of the coupons match these changes. Moreover, the measurements were burdened with large deviations. This can be partly explained by the unevenness of the color change on and the measurement of different parts of the coupons; even after 30 months of exposure, most surfaces did not show a uniform appearance, only local changes. Nevertheless, it may also be a consequence of some procedural errors, as indicated by the opposite trend of the total color difference in location 6, which does not reflect reality (Figure 6).

The problem with assessing zinc color changes (Figure 7) is that they are small and not visible at first glance, even on the coupons. Only under a stereomicroscope were they found locally (Figure 8). After including the error bars, most of the dependencies disappeared, however the biggest changes seemed to occur in churches (locations 14–16), i.e., under conditions of higher relative humidity.

For lead, two trends were evident (Figure 9). Most lead coupons rapidly changed color in the first months of exposures as their surfaces darkened; thereafter, the percentage increase in change was not as significant. However, this was not the case for coupons from churches (locations 14–16) and depositories (locations 8–10), where the changes had a growing trend for most of the period under review, probably due to the low content of acetic acid in the air. As with silver and copper coupons, the greatest color changes were not associated with the highest concentrations of acetic acid. Comparing to those metals, the reason is not a low sensitivity of lead to acetic acid, but a light shade of corrosion products (Figure 10).

### 3.2. Mass Loss of Corrosion Coupons

Compared to the mass increase, the mass loss should better reflect the reality of the aggressiveness of an environment, because it is due to the loss of metal and its conversion into corrosion products, which are removed after exposure by pickling. However, since an evaluation according to the ISO standard 11844 is provided by comparison with pickling of an unexposed sample, the pickling bath must not be too aggressive. Otherwise, the results will be burdened with a large error, where the mass loss of the blank specimen may completely or largely hide the mass changes caused by exposure, so that the corrosion is then underestimated. It could have greatly influenced the results of the three-month exposures of silver and copper coupons from a low-corrosive environment. When silver coupons of 1 × 1 cm in size exposed for only 3 months were pickled, the mass loss of blank unexposed samples was 50 mg∙m^−2^. Compared to corrosion rates in Figure 11, it is evident that the loss of the unexposed blank specimen represented a non-negligible fraction of the total mass loss, sometimes even resulting in a negative value after subtracting the blank loss. A similar behavior was also seen in the case of copper (Figure 12), which may explain the higher mass loss after one-year exposures in locations 1–5 than after short-term exposures.

Despite the aggressiveness of the pickling bath, the mass loss results of silver and copper coupons confirmed the low sensitivity of copper and silver to environmental contamination by acetic acid vapors, as already indicated by color changes. Relative humidity and other air pollutants had a larger effect because corrosion rates were the highest in location 15, which most closely corresponds to the outdoor environment with air circulation, humidity, and temperature fluctuations, but with a low concentration of acetic acid. Conversely, the corrosion rate in locations with a high content of acetic acid (locations 11, 12, or 18) were lower or comparable to those with a many times lower emission of the acid into the environment.

Corrosion rates of zinc based on the mass loss showed two trends. Firstly, zinc was more sensitive to acetic acid pollution compared to silver and copper, as illustrated by location 18 in Figure 13. Secondly, zinc was less resistant to an environment with a higher relative humidity (represented by locations 14–16). This is the difference in the corrosion behavior of zinc and lead. Lead is particularly sensitive to the content of volatile organic acids, as seen in Figure 14; thus, the highest corrosion rates correspond to the locations with the highest acetic acid concentration (locations 4, 11, 12, and 18), and similarly, low corrosion rates were detected in low acetic acid environments, regardless of humidity values (e.g., comparison of locations 8 and 15). Moreover, a linear relationship between the acetic acid concentration and the corrosion rate calculated from mass loss existed and is shown in Figure 15. The figure compares the annual values and illustrates two linear regressions with very satisfactory goodness of fit (the R^2^ coefficients of 0.66 and 0.80), where the values of location 13 were omitted in the second case to improve the match. It is not clear why the data from the location did not fit into the dependence, it could be a combination of low relative humidity, not too high acid concentration, and limited sources of other pollutants, because it is a depository. Linear relationships between corrosion and acetic acid concentration have been obtained in other published studies, too, although the equations have different slopes and coefficients [20,21,22,23]. These studies mainly focus on the relationship between the corrosion rate, acid concentration, and relative humidity. In contrast, our localities represent conditions where corrosion rates are determined not only by humidity content, but also by temperature and pollutant concentration, or by variations in conditions depending on the season and time of day. Nevertheless, over the wide range of test conditions to which the collection objects may be exposed, the dependence of lead’s corrosion rate on the acetic acid concentration appears to be linear. However, the annual mass loss and annual average concentration of acetic acid are better to compare, because the amount of acid varies in the air during a year, as well as the corrosion rate.

## 4. Discussion

The color changes of the lead coupons have confirmed that the most rapid changes in color occurred in the first weeks or months of exposure and are linked to darkening of the lead surface, with the rate of darkening influenced by the aggressiveness of the environment. Therefore, with an increasing time of exposure, lead coupons have much more time to darken, even in a low corrosive environment, causing the colorimetric differences to blur. This is then reflected in the comparison of color results with corrosion rates in Figure 16, which illustrates a linear relationship between the corrosion rate based on mass loss after 1-year exposure and the total color difference of lead samples after different times of exposure. Corrosion rates from one-year exposure matched the color changes after three months and a month, respectively, better (represented with the higher value of the coefficient of determination) than after a year, although there was a smaller dataset for comparison. In addition, the regression slope decreased with the increasing exposure time, also indicating a slowing down of color changes. However, extreme conditions in terms of a high acetic acid concentration should be excluded from the dependence due to the formation of colored corrosion products, especially of light shades, reducing the effect of darkening (represented by a shift of the parameter L to lower values), which was dominant in the case of the total color change of lead coupons, as shown in Figure 17. Results of three-month exposures were also compared with each other, but no clear dependence was found, probably because the evaluation of color and mass changes was carried out on different sets of samples exposed at different times of the year, i.e., under various values of temperature and relative humidity, parameters that strongly influence corrosion rates.

Darkening of coupons plays an important role in the process of color changes of silver, too, but an effect of other parameters becomes apparent as well. In many cases of silver coupons (Figure 18), the parameter b, which corresponds to the blue–yellow color transition, is dominant or at least comparable to darkening. A consequence of the more complex color change of silver is probably the lower agreement of color and mass measurements compared to lead. However, a linear relationship between the methods was evident (Figure 19) and very satisfactory for results of silver coupons exposed for a year and for 30 months. The comparison of corrosion rates after one year of exposure with color measurements after a shorter exposure showed that the agreement of the results improved with the increasing exposure time of silver coupons. Nevertheless, the color changes after only three months of exposure were sufficient to correspond well to the annual mass changes. A similar trend can also be observed when comparing the results of equally long exposures. As the exposure time increased, the goodness of fit improved, and in addition, the slope of the line became steeper. The results are consistent with the trend of color changes of silver, where the total color differences between short- and long-term exposures were significant.

In the case of copper coupons, on the one hand, the parameter b had as significant an influence on color change as the parameter L, but on the other hand, the third parameter was not negligible either (Figure 20). A copper surface can then take on different shades and colors and is not uniform, which complicates color measurements and can cause large measurement deviations, such as in the first months of copper exposure in Figure 12. Combined with the aggressiveness of the pickling solution, which caused underestimation of corrosion, there was little comparison of or no correlation between the mass loss and the color change (Figure 21). The best fit was achieved for annual exposures, but the value of the R^2^ coefficient (0.33) was still low for any conclusions.

Similar to copper, the comparison of the methods did not indicate any important trend (Figure 22) for zinc as the slopes of the lines were too small and the R^2^ coefficients were as well. There seems to be no relationship between the methods. The explanation probably lies in the small color differences, reflecting a typical greyish or whitish color of corrosion products of silver-grey zinc; thus, they did not cause a large shift of the L parameter from its original value, which was responsible for most of the color changes of the zinc coupons (Figure 23).

## 5. Conclusions

Colorimetric measurement can by no means fully replace a measurement of climatic parameters, determination of pollutant concentrations, or an assessment of corrosion aggressiveness based on mass loss, i.e., methods described in ISO standards for classification of environmental corrosiveness, because it is too influenced by uniformity of color change and the nature of corrosion products. Even so, the comparison of the color measurement with the mass changes showed that there was a dependence between the methods for silver and lead. This suggests the possibility of using the color measurement as a simple tool for estimating the aggressiveness of an indoor environment in a significantly shorter time. The method can be particularly useful in assessing the risk of volatile organic acids, to which lead is very sensitive, and if the concentration does not exceed units of ppm, the results of color changes after a few weeks of exposure correlate well with mass changes.

## Figures and Tables

**Figure 1 materials-16-00226-f001:**
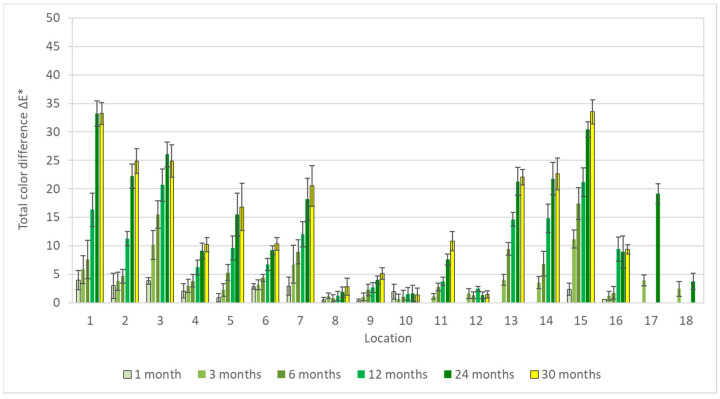
Total color differences of silver corrosion coupons after 1–30 months of exposure.

**Figure 2 materials-16-00226-f002:**
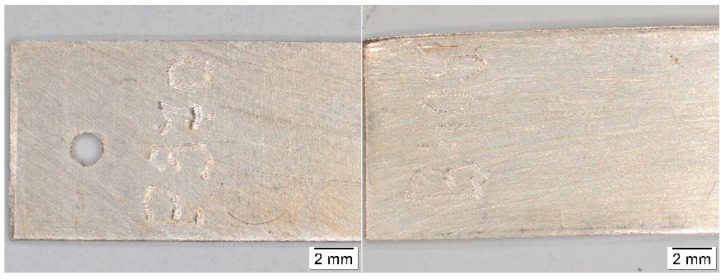
Comparison of the appearance of silver coupons after exposure in an archive (**left**) and in a wooden box with an acetate film (**right**).

**Figure 3 materials-16-00226-f003:**
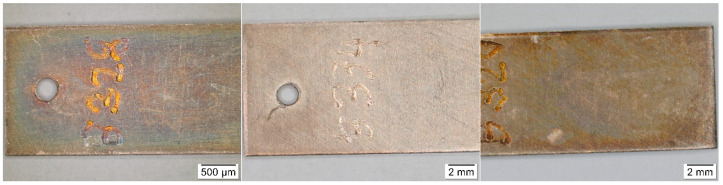
Comparison of the appearance of silver coupons stored in an open area of a library (**left**), in a wooden cabinet (**middle**), and in a church (**right**).

**Figure 4 materials-16-00226-f004:**
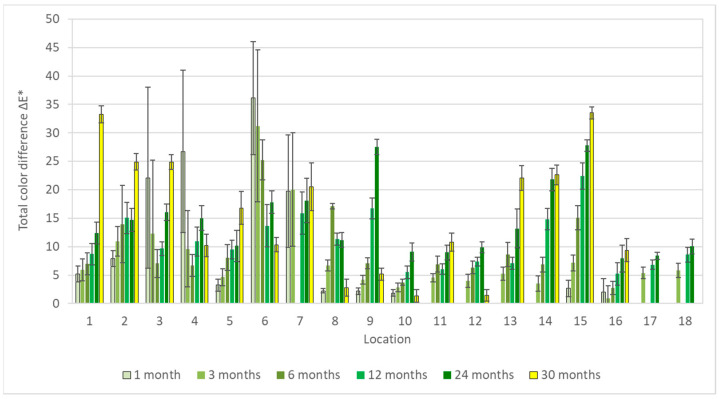
Total color difference of copper corrosion coupons after 1–30 months of exposure.

**Figure 5 materials-16-00226-f005:**
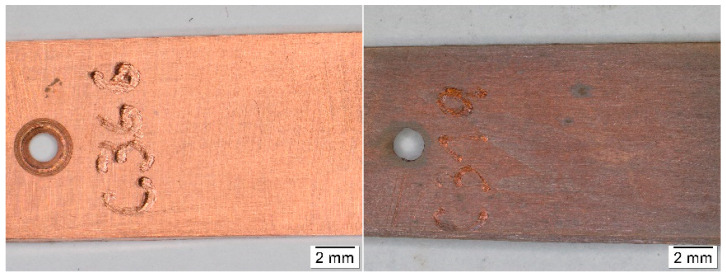
Comparison of the appearance of the copper coupons with the highest value of the total color difference (a library from location 1 on the (**left**), a church from location 15 on the (**right**)).

**Figure 6 materials-16-00226-f006:**
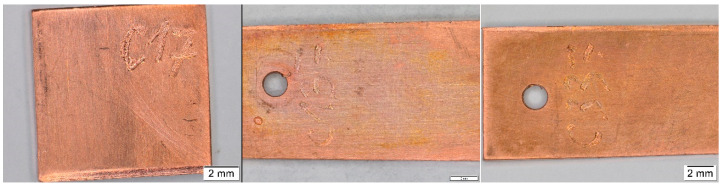
Comparison of the appearance of the copper coupons from location 7 (depository) after 3 months (**left**), 12 months (**middle**), and after 30 months (**right**).

**Figure 7 materials-16-00226-f007:**
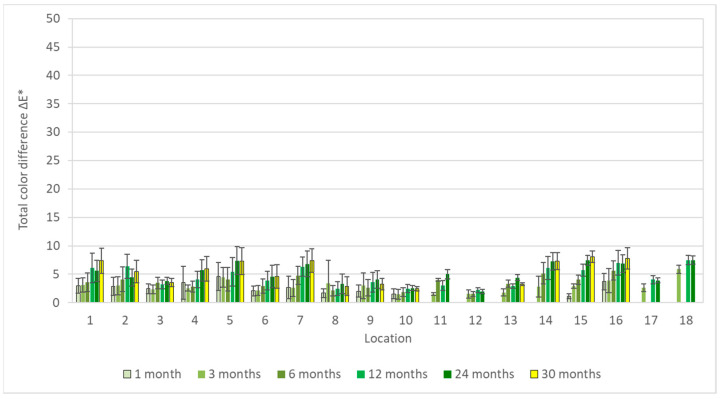
Total color difference of zinc corrosion coupons after 1, 3, and 30 months of exposure.

**Figure 8 materials-16-00226-f008:**
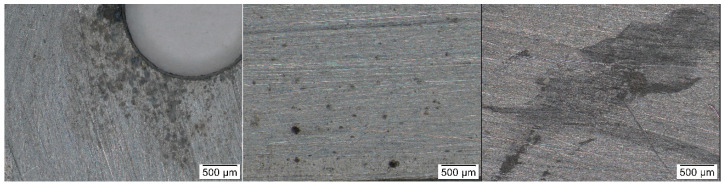
Details of zinc coupons from a church of location 15 (**left**), a wooden box with acetate film (**middle**), and a library from location 5 (**right**).

**Figure 9 materials-16-00226-f009:**
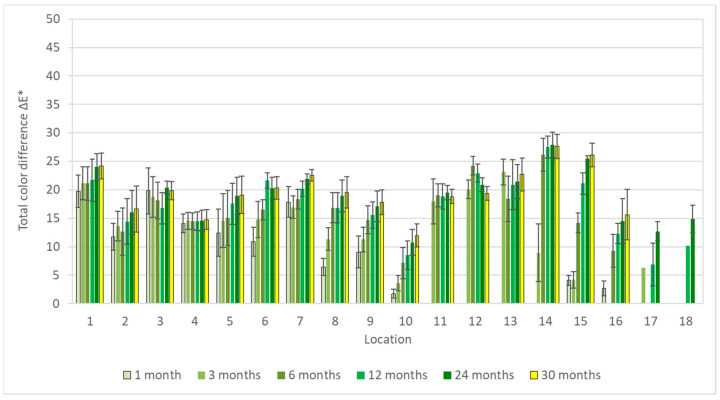
Total color difference of lead corrosion coupons after 1–30 months of exposure.

**Figure 10 materials-16-00226-f010:**
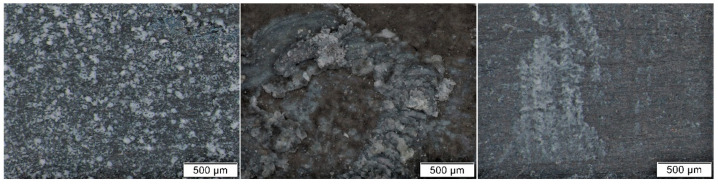
Details of lead coupons from a wooden showcase, a case with an acetate film, and a wooden box with an acetate film (locations 11, 12, and 18) after 30-month exposure.

**Figure 11 materials-16-00226-f011:**
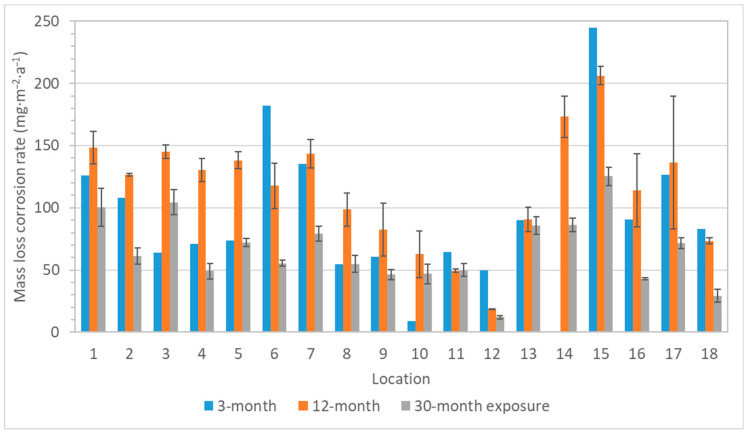
Corrosion rate calculated from mass loss of silver coupons (error bars for 3-month exposure are missing due to the small number of exposed samples).

**Figure 12 materials-16-00226-f012:**
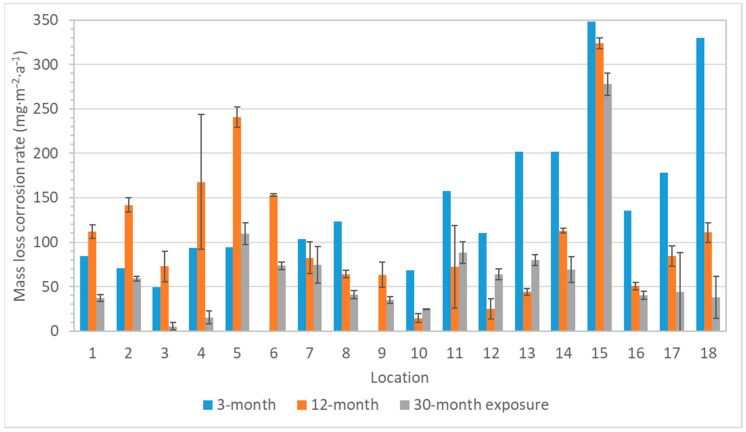
Corrosion rate calculated from mass loss of copper coupons (error bars for 3-month exposure are missing due to the small number of exposed samples).

**Figure 13 materials-16-00226-f013:**
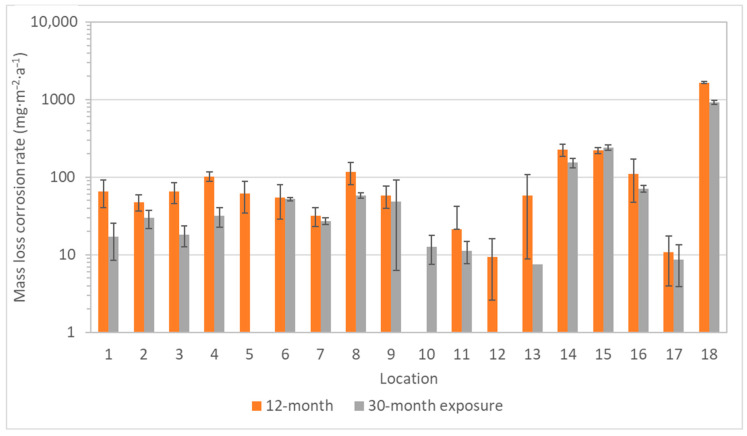
Corrosion rate calculated from mass loss of zinc coupons (logarithmic *y*-axis scale, 3-month exposure of zinc coupons did not take place).

**Figure 14 materials-16-00226-f014:**
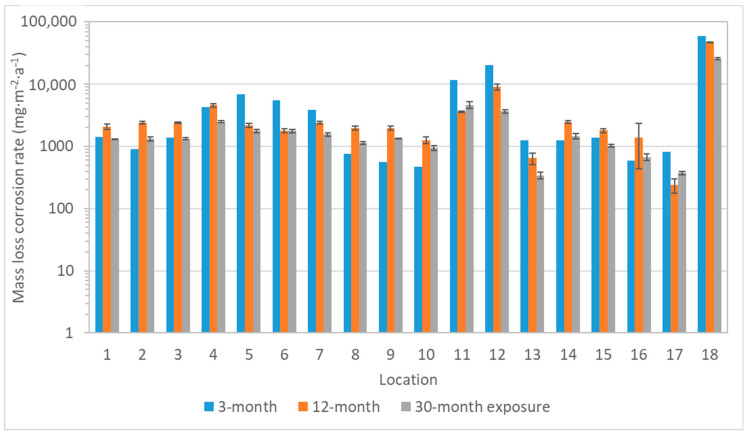
Corrosion rate calculated from mass loss of lead coupons (logarithmic *y*-axis scale, error bars for 3-month exposure are missing due to the small number of exposed samples).

**Figure 15 materials-16-00226-f015:**
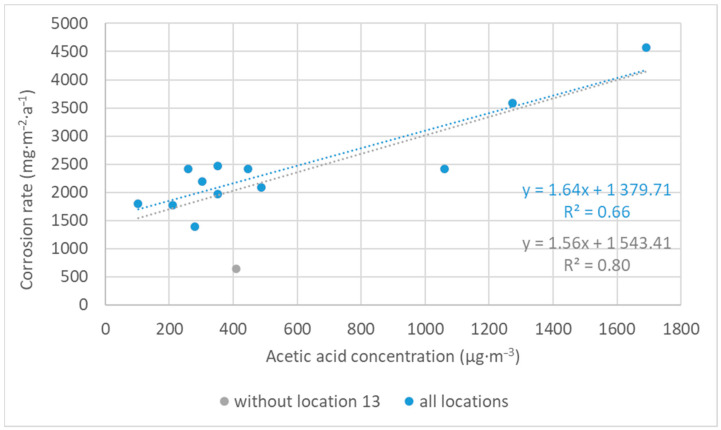
Dependence of corrosion rate of lead based on mass loss on acetic acid content in the air.

**Figure 16 materials-16-00226-f016:**
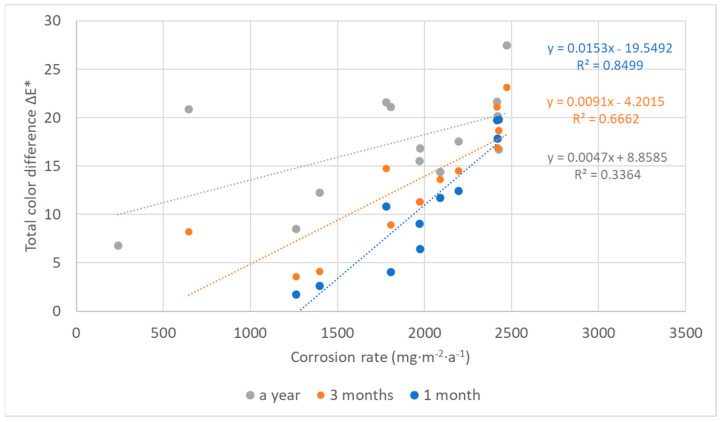
Comparison of corrosion rates of lead based on mass losses after one-year exposure with total color changes after 1, 3, and 12 months (locations 4, 11, 12, and 18 are excluded).

**Figure 17 materials-16-00226-f017:**
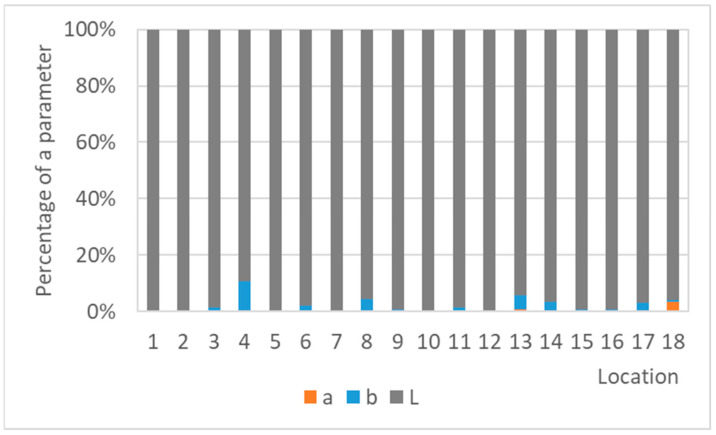
Percentage of the color parameters in the total color difference of lead coupons exposed for a year.

**Figure 18 materials-16-00226-f018:**
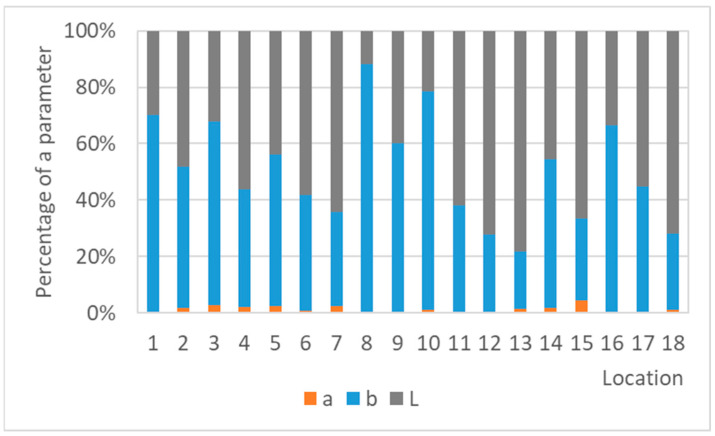
Percentage of the color parameters in the total color difference of silver coupons exposed for a year.

**Figure 19 materials-16-00226-f019:**
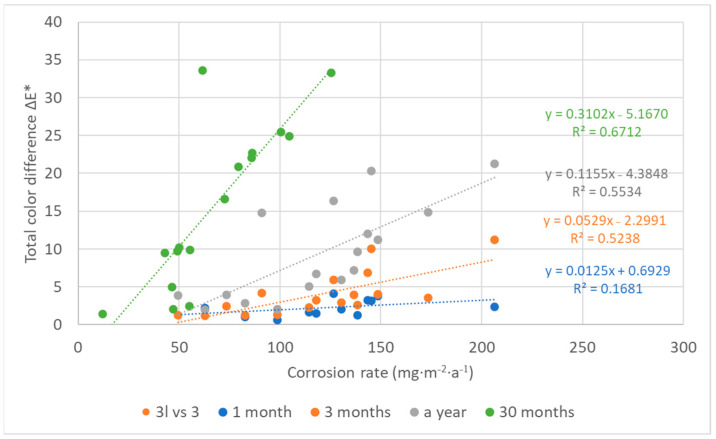
Comparison of corrosion rates of silver based on mass losses after 1-year exposure with total color changes after 1, 3, and 12 months, and mass losses after 30 months with color changes after the same time.

**Figure 20 materials-16-00226-f020:**
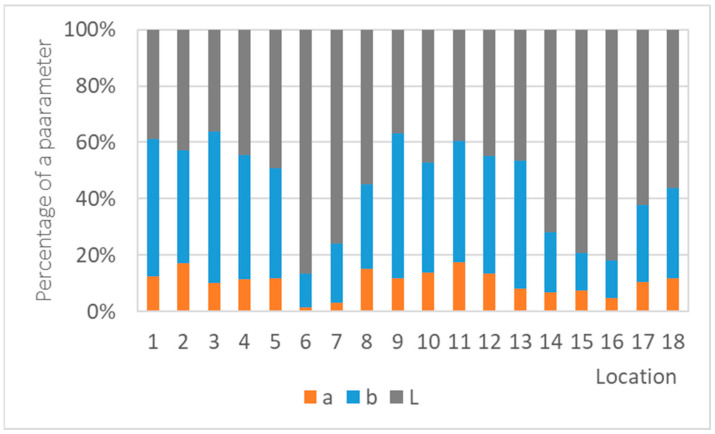
Percentage of the color parameters in the total color difference of copper coupons exposed for a year.

**Figure 21 materials-16-00226-f021:**
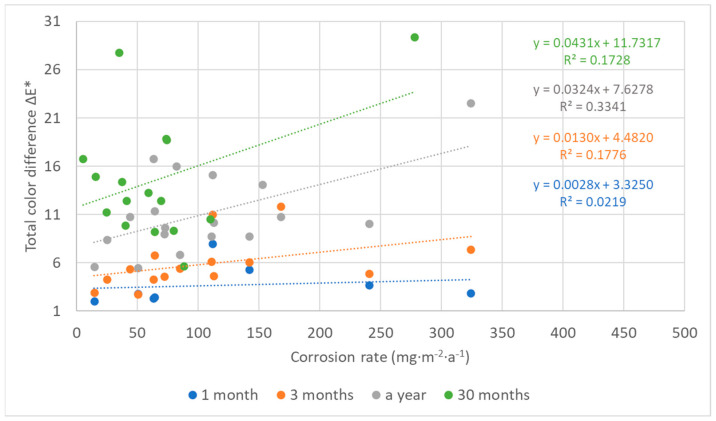
Comparison of corrosion rates of copper based on mass losses after 1-year exposure with total color changes after 1, 3, and 12 months, and mass losses after 30 months with color changes after the same time (locations 3, 4, 6, and 7 were excluded in the case of 1-month and 3-month exposures).

**Figure 22 materials-16-00226-f022:**
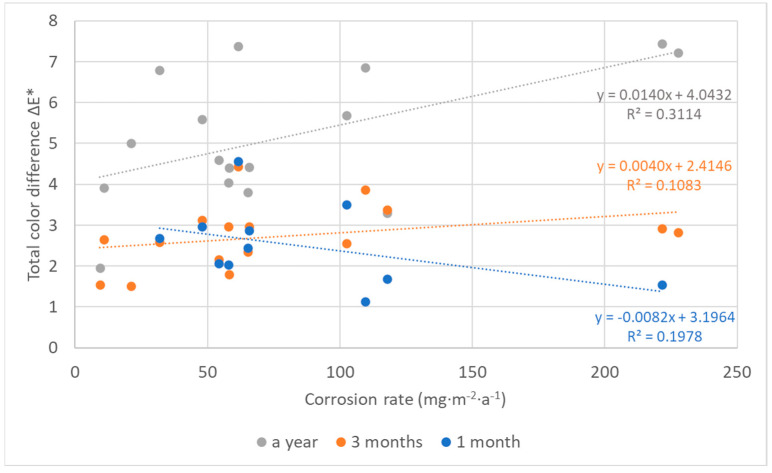
Comparison of corrosion rates of zinc based on mass losses after 1-year exposure with total color changes after 1, 3, and 12 months (location 18 was excluded).

**Figure 23 materials-16-00226-f023:**
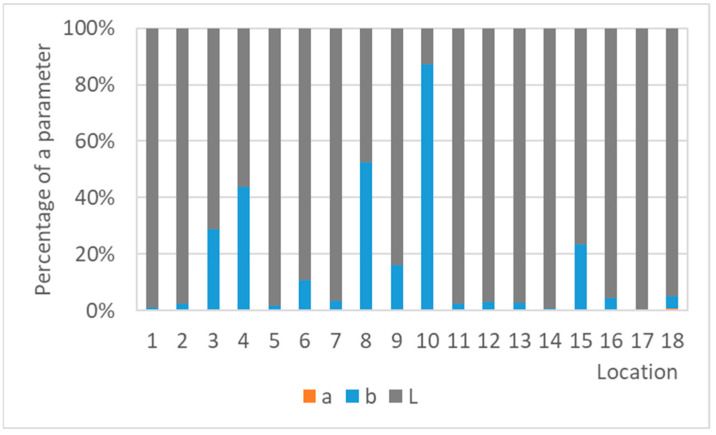
Percentage of the color parameters in the total color difference of zinc coupons exposed for a year.

**Table 1 materials-16-00226-t001:** Values of temperature, relative humidity, and acetic acid in the locations.

Location	Temperature (°C)			Relative Humidity (%)			Acetic Acid Content (µg∙m^−3^)		
	Avg	Min	Max	Avg	Min	Max	Avg	Min	Max
1—library	20.9	16.5	28.9	39.7	21.6	67.0	488	139	965
2—showcase in library	21.2	17.6	28.6	38.7	29.9	47.9	1061	356	2324
3—exhibition hall	20.3	15.8	30.0	41.4	23.2	63.6	446	221	1113
4—showcase in exhibition hall	20.3	15.4	30.2	39.8	30.6	49.6	1690	891	3241
5—library ^1^	18.3	11.2	26.5	50.4	45.6	55.7	302	132	570
6—depository	20.4	17.9	26.2	49.4	35.3	60.1	351	248	444
7—depository	19.0	14.3	27.5	49.9	36.1	62.1	259 ^5^	212 ^5^	287 ^5^
8—archive	15.8	14.0	17.7	51.3	47.0	56.4	211 ^5^	123 ^5^	261 ^5^
11—showcase in exhibition hall	25.0	19.4	28.9	31.5	18.0	46.9	1272	487	2581
12—case with acetate film	23.0	21.0	25.9	40.7	30.5	52.6			
13—depository ^2^	22.2	20.1	24.5	42.4	26.7	57.6	409	192	524
14—church	12.8	-2.1	24.6	65.1	46.6	88.3	351	201	581
15—church	10.5	-3.7	25.6	75.2	38.2	99.9	102 ^5^	41 ^5^	300 ^5^
16—church ^3^	10.7	-0.3	24.3	69.3	47.4	95.4	280	204	349
17—archive ^4^	15.0	14.1	17.6	50.8	42.2	57.9			

Abbreviations: Avg, Min, and Max mean annual average, minimum, and maximum in the year. Missing data of temperature and humidity: ^1^ from April to June, ^2^ from November to January, ^3^ September, and ^4^ from June to September. The number ^5^ indicates measurements that took place only during a certain part of the year.

## Data Availability

Not applicable.

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
