# Peer review of "Color Measurement in the Corrosivity Assessment of Museums, Archives, and Churches"

_materials, 2022, doi:10.3390/ma16010226_

Round 1

Reviewer 1 Report

This paper reports on the use of colorimetric measurements as a quick and simple tool for studying material corrosion. To evaluate the potentiality of the method they compare data from colorimetry with standard corrosion investigations (mass loss of metallic coupons) following the ISO-11844 standard.

The idea is interesting, the comparison between both methods is well driven and the paper reads well. Therefore, I recommend its publication in materials. I have only few suggestions for improving the reading and understanding of the paper:

1/ Line 62 - For the first group, I would list the corresponding locations in brackets.

2/ Line 202 (figure 10) - Images are a bit dark. if it is possible to lighten them, it would be better.

3/ Line 242, it is necessary to put more details on what is seen on figure 15. Is a linear variation expected? Also, the removal of location 13 for fitting is not well explained.

4/ Lines 255-259. It would be good to explain how is constructed figure 16. In particular it is not clear why the corrosion rate after one year is compared with the colorimetric measurements performed after shorter periods. Also, what is the evaluation criterium? (value of R2?)

5/Line 275, 293, 303. It would be good to give a more detailed description of figures 19, 21 and 22.

6/ Figure 22, better to replace 1-12 months by 1, 3 and 12 months as for previous figures.

Author Response

Thank you for your revision. I hope I've incorporated all the comments, as you can see in the attached document. 

Reviewer 2 Report

The paper is dealing with comparing corrosion rate and change of colour measurements in corrosive environment containing acetic acid for metallic coupons of silver, copper, zinc and lead. The paper is clearly written with plenty of the results. It was found a good correlation between the methods for silver and lead, confirming the possibility of non-destructive monitoring of corrosion by measurement of colour changes.

Comments:

Formal: The sentence line 32/33 is unfinished. Line 82 Error sentence in bold is inside. Table 1: Are maximal values of acetic acid in the correct units?

2.1. Corrosion experiment

2.2. Preparation of coupons and their exposure

What was the reason to evaluate the samples under 1 year exposure when the standard itself recommends 1 year exposure?

2.3. Evaluation methods

Can you be more specific in describing neglecting the edges (line 123) in calculation of corrosion rates of the coupons?

3. Results

Recommendations:

Concerning graphs, I suggest applying different colour than yellow due to difficulty with reading the values in the printed form. The recommendation is not mandatory. Fig 2, 5, 8, 10 – instead of using numbers under the figures I suggest name the locality. Chapter 3.2. – although it is clearly explained in the text I would recommend to add sentence why there are not error bars for 3 months exposure in Fig. 11, 12 and 14 and why the results for zinc coupons after 3 months exposure are missing.

4. Discussion

It would be appropriate to specify that short-term measurements after Pb exposure and long-term after Ag exposure provide good correlation between the measurements. Namely in the case of Ag one month exposure of Ag to corrosive environment the dependency shows R2 = 0.1681 and in the case of Cu after 3- and 30-months exposure dependency with R2 about 0.17 are differently interpreted. Concerning Ag dependance is interpreted in a positive way and in the second case in a negative way.

Conclusion:

Due to several formal mistakes I recommend publishing the paper after minor revision.

Author Response

Thank you for your revision. I hope I have incorporated all the comments, as you can see in the attached document. The changes are visible in red in the revision mode. 
